Transcriptome analysis of Citrus Aurantium L. to study synephrine biosynthesis during developmental stages

Zhong Can 1 2 3
Yang Xitao 1
Niu Juan 1
Zhou Xin 2
Zhou Jiahao 1
Pan Gen 1 2
Sun Zhimin 1
Chen Jianhua 1
Cao Ke 3
Luan Mingbao luanmingbao@caas.cn 1 4
1 Institute of Bast Fiber Crops, Chinese Academy of Agricultural Sciences, Key Laboratory of Stem-Fiber Biomass and Engineering Microbiology, Ministry of Agriculture , Changsha , Hunan , China
2 Institute of Chinese Medicine Resources, Hunan Academy of Chinese Medicine , Changsha , Hunan , China
3 The Key Laboratory of Biology and Genetic Improvement of Horticultural Crops (Fruit Tree Breeding Technology), Ministry of Agriculture, Zhengzhou Fruit Research Institute, Chinese Academy of Agricultural Sciences , Zhenzhou , Henan , China
4 National Nanfan Research Institute, Chinese Academy of Agricultural Sciences , Sanya , Hainan , China
Singh Anshuman
Electronic publication date: 2024 Sep 9
Publication date: 2024
Volume: 12
Electronic Location ID: e17965
Received 2024 Jan 26; Accepted 2024 Aug 2
Copyright: ©2024 Zhong et al.
Copyright year: 2024
Copyright holder: Zhong et al.
License: This is an open access article distributed under the terms of the Creative Commons Attribution License, which permits unrestricted use, distribution, reproduction and adaptation in any medium and for any purpose provided that it is properly attributed. For attribution, the original author(s), title, publication source (PeerJ) and either DOI or URL of the article must be cited.
License URL: https://creativecommons.org/licenses/by/4.0/

Keywords: Aurantii Fructus Immaturus, Synephrine, Alkaloids, TyDC

Funding: The Agricultural Science and Technology Innovation Program (ASTIP) of CAAS 2017IBFC The Central Public-interest Scientific Institution Basal Research Fund 1610242020010 This work was supported by the Agricultural Science and Technology Innovation Program (ASTIP) of CAAS (grant number 2017IBFC) and the Central Public-interest Scientific Institution Basal Research Fund (grant number 1610242020010). The funders had no role in study design, data collection and analysis, decision to publish, or preparation of the manuscript.

==============================
Citrus aurantium L., sometimes known as “sour orange,” is an important Chinese herb with young, immature fruits, or “zhishi,” that are high in synephrine. Synephrine is a commonly utilized natural chemical with promising applications in effectively increasing metabolism, heat expenditure, energy level, oxidative fat, and weight loss. However, little is known about the genes and pathways involved in synephrine production during the critical developmental stages of C. aurantium L., which limits the development of the industry. According to this study, the concentration of synephrine gradually decreased as the fruit developed. Transcriptome sequencing was used to examine the DEGs associated with synephrine connections and served as the foundation for creating synephrine-rich C. aurantium L. Comparisons conducted between different developmental stages to obtain DEGs, and the number of DEGs varied from 690 to 3,019. Tyrosine and tryptophan biosynthesis, glycolysis/gluconeogenesis, pentose phosphate pathway, phenylalanine, and tyrosine metabolism were the main KEGG pathways that were substantially enriched. The results showed that 25 genes among these KEGG pathways may be related to synephrine synthesis. The WGCNA and one-way ANOVA analysis adoption variance across the groups suggested that 11 genes might play a crucial role in synephrine synthesis and should therefore be further analyzed. We also selected six DEGs at random and analyzed their expression levels by RT-qPCR, and high repeatability and reliability were demonstrated by our finished RNA-seq study results. These results may be useful in selecting or modifying genes to increase the quantity of synephrine in sour oranges.

Introduction

The Rutaceae family member sour orange (Citrus aurantium L.) is thought to be native to northeastern India, northern Burma, and southern China (Rao, Zuo & Xu, 2021). The unripe sour orange fruit, Aurantii Fructus Immaturus (AFI), also known as zhishi in Chinese, has been utilized for thousands of years as a popular traditional edible-medicinal plant with regulatory visceral activities in many herbal formulae due to its strong medicinal properties (Liu et al., 2017). Dried immature fruits are usually picked in May or June of each year, much like those of Citrus sinensis Osbeck (often known as sweet oranges), Citrus aurantium L. (sometimes known as bitter oranges), and other cultivars like C. junos, C. sjaponica, and C. bergamia (Deshmukh et al., 2017). AFI effectively treats gouty arthritis, cancer, and cardiovascular disorders when administered alone to eliminate phlegm (Wu et al., 2020). C. aurantium mainly contains alkaloids, flavonoids, glycosides, and volatile oil compounds, with the aromatic alkaloid synephrine attracting much attention due to its high specificity as a quality marker of C. aurantium (Bader et al., 2017; Pellati & Benvenuti, 2007).

In the food, beverage, pharmaceutical, and healthcare sectors, synephrine is a commonly utilized natural chemical with promising applications. According to a previous survey, synephrine can effectively increase metabolism, heat expenditure, energy level, oxidative fat, and weight loss, and is clinically used to treat bronchial asthma, hypotension, prostration, shock, and postural hypotension that occur during surgery and when under anesthesia (Haaz et al., 2006). When used as an ephedrine substitute in dietary supplements, synephrine exerts a better weight-loss effects (Sander et al., 2005) and holds great potential for treating weight-gain-related diseases (Weil, Qunz & Yang, 2016). Compared with other citrus species, C. aurantium has the highest synephrine content in its basal source plant, sour orange (Zhang, 2019), and accounts for 90% of sour orange’s total raw alkaloid content. Therefore, C. aurantium is the main raw material used for synephrine extract. Synephrine is found in sour orange peel, pulp and leaves. However, the synephrine content in sour oranges is unstable and decreases as the fruit matures (Bai et al., 2018; Shi et al., 2019). Consequently, it is important to understand how plants produce synephrine and enhance citrus trees’ synephrine content through breeding.

The synephrine biosynthesis pathway in plants is a branch of the tyrosine pathway (Stohs & Preuss, 2012). In the first reaction step of the pathway, phosphoenolpyruvate (PEP) from the glycolysis/gluconeogenesis pathway and erythrose-4-phosphate (E4P) from the pentose phosphate pathway are converted to 3-deoxy-d-arabinoheptanoate-7-phosphate for the synthesis of aromatic amino acids via the shikimate pathway by DAHPS synthase (Arbona et al., 2013). The synthesis component chorismite, an important hub product, is catalyzed continuously by several enzymes, and part of the catalyzed chorismite is used to synthesize L-tryptophan and L-phenylalanine (Liu et al., 2021). The other moiety generates p-hydroxyphenylpyruvate (HPP) catalyzed by the chorismate mutase/prephenate dehydrogenase bifunctional enzyme Tyra, which ultimately generates tyrosine through interaction with transamination (De Oliveira et al., 2019). Tyrosine can also be produced through arogenate dehydrogenase (tyraat)-mediated catalysis of prephenate-derived L-arogenate. Both pathways are present in C. aurantium. Tyrosine is then further catalyzed by aromatic-l-amino acid decarboxylase (DDC) or tyrosine decarboxylase (TyDC) to produce the phenethylamine-type alkaloid tyramine, which in turn synthesizes alkaloids (Wang et al., 2022).

The plant material used in this work to examine the transcription mechanisms during the fruit development stage was C. aurantium L. (C17). The fruits were harvested in eight stages: sample C17_1 was harvested at 45 days after full blooming (DAFB), followed by samples C17_2, C17_3, C17_4, C17_5, C17_6, C17_7, and C17_8 harvested every 15 days until 150 DAFB. According to our analysis of their physiological characteristics, the samples’ synephrine levels dropped as the growth interval increased. Significant differences in the synephrine content existed between the C17_1, C17_4, and C17_7 stages. Finally, samples from these three stages were selected for transcriptome analysis. The primary objectives of the study were to (1) determine the change rule of synephrine content in AFI, (2) search for genes associated with synephrine biosynthesis that were expressed differently, and (3) build a correlation network for synephrine biosynthesis and important enzyme-encoding genes in the process. The objective of the work was to establish a basis for comprehending the molecular mechanisms behind C. aurantium L.’s synephrine production.

Materials and Methods

Plant sample collection

C. aurantium L. was cultivated at the Chinese Academy of Agricultural Sciences’ experimental area for bast fiber crops in Yuanjiang, Hunan province. The distinct growth phases of C17 were chosen for study because of the notable variations in synephrine levels between the stages. Samples were taken between May and September in 2020 and 2021during the fruit growth stage. Fruits were harvested at 45 DAFB, and then every 15 days until 150 DAFB. In each stage, fruits from C17 were pooled to produce three biological replicates. The obtained fruits were divided in half. One half was used to analyze the synephrine content, and the other half was quickly frozen in liquid nitrogen and stored in an ultra-low freezer (−80 °C) for transcriptome sequencing.

Synephrine content analysis

Fruits were harvested at 45 (C17_1), 60 (C17_2), 75 (C17_3), 90 (C17_4), 105 (C17_5), 120 (C17_6), 135 (C17_7), and 150 (C17_8) DAFB. They were then crushed into powder and dried at 60 °C to a constant weight. We mixed 1 g powder with 30 mL of absolute methanol and incubated the mixture at 70 °C for 2 h to extract synephrine via the reflux extraction. A high-speed centrifuge was used to centrifuge the extraction solution for 10 min at 3,000 rpm. The supernatant was then filtered through a 0.45 µm filter membrane. The supernatant was then analyzed using high-performance capillary electrophoresis (HPCE, Agilent 7100; Agilent, Santa Clara, CA, USA) with 60 mM sodium tetraborate-potassium dihydrogen phosphate and 30% methanol as the mobile phase at a pressure injection of 50 mbar × 5 s and temperature of 25 °C. The synephrine content was ascertained using an external standard approach by taking the consistence as the Y-axis, the peak area as the X-axis, and the standard curve was drawn (y = 410.87x −3.0689) (Lyu et al., 2022). Two technical and three biological replicates were performed in this study.

cDNA library construction, sequencing and analysis

Using the RNA prep Pure Plant kit, total RNA was extracted from three accessions (three developmental stages): 45 DAFB (C17_1_1, C17_1_2, and C17_1_3), 90 DAFB (C17_4_1, C17_4_2, and C17_4_3), and 135 DAFB (C17_7_1, C17_7_2, and C17_7_3) (Tiangen, Beijing, China). Thermo Fisher Scientific (Waltham, Massachusetts, MA, USA) provided a NanoDrop Agilent 2100 bioanalyzer that was used to measure the concentration and purity of RNA. Nine cDNA libraries for each of the three fruits were created in preparation for RNA sequencing (each stage comprised three duplicates). In short, the NEBNext Ultra RNA Library Prep Kit (Illumina; NEB, Ipswich, MA, USA) produced the cDNA libraries after mRNA was separated from 1 ug of total RNA. The system known as the Agilent Bioanalyzer 2100 (Agilent Technologies, Palo Alto, CA, USA) was utilized to evaluate the quality of the cDNA library, and DNBseq employed the Phred +33 quality system to carry out the Illumina sequencing. The SOAPnuke program was used to exclude low-quality readings and those that containedpoly-N (Chen et al., 2018). The remaining clean reads were aligned to the reference genome of the C. sinensis genome (Citrus sinensis v3.0, database: GCA_018105775.1) using hierarchical indexing for spliced alignment of transcripts 2 (HISAT2 (v2.1.0)) (Kim et al., 2019) with orientation mode to obtain unigenes. Based on the selected reference genome sequence, the mapped reads were spliced by StringTie (http://ccb.jhu.edu/software/stringtie/) to find the original unannotated transcripts and new genes of the species to supply and improve the original genome annotation information. The transcripts without corresponding templates in the reference genome annotation information were defined as new transcripts and new genes.

Differentially expressed genes and functional enrichment

We measured each transcript’s expression level using the transcripts per million reads (TPM) method to find the differentially expressed genes (DEGs) between the two samples. Gene abundances were measured using RNA-Seq by Expectation Maximization (RSEM) (Li & Dewey, 2011), and differential expression analysis was carried out using DESeq2 (Love, Huber & Anders, 2014). DEGs were considered significantly distinct if they had —log2FC— ≥1 and FDR ≤ 0.05 (DESeq2). PCA and hierarchical clustering heat map correlation were analyzed in network (https://www.majorbio.com) among samples based on FPKM of all unigenes. Functional enrichment analysis was also carried out using KEGG (Kanehisa & Goto, 2000) to find the significantly enriched DEGs in metabolic pathways at a Bonferroni-corrected P-value ≤0.05 relative to the background of the entire transcriptome. KEGG pathway enrichment analysis was performed using KOBAS (Xie et al., 2011) on the Majorbio platform (https://www.majorbio.com).

The construction of co-expression networks and weighted correlation network analysis

The Majorbio platform was utilized to create a weighted gene co-expression network (https://www.majorbio.com). An unsigned gene co-expression network was built using 7,309 DEGs chosen from the three accessions’ DEGs for weighted correlation network analysis (WGCNA). The sample preprocessing parameters are TPM mean = 1 and coefficient of variation = 0.1. Sample cluster analysis was performed to identify and remove outlier samples. The results showed that the samples were clustered into nine categories. Genes were divided into modules based on their expression trends, and the default parameter was selected as networkType = signed, soft Power(β-Value) = 9, minModuleSize = 30, minKMEtoStay = 0.3, mergeCutHeight = 0.25. We assigned 6,072 genes to a total of eight modules, with the grey module representing genes that were not assigned to specific modules and was therefore excluded.

RT-qPCR analysis

Reverse transcription PCR with quantitative analysis was used to validate the RNAseq data (RT-qPCR). The total RNA extracted from nine samples (three duplicates of each of the three stages) was reverse transcribed to cDNA using PrimeScript RT Master Mix for qPCR (Takara Biotechnology Co., Ltd., Dalian, China). A primer set was designed based on each identified gene sequence of the transcriptome library using Primer Premier 5.0 (Table S7). The Step One Plus Real-Time PCR Systems was used to evaluate the relative expression of proposed pathway genes in studied samples with Power SYBR® Green I Master Mix (Applied Biosystems, Waltham, MA, USA). Prerequisite parameters were set as initial denaturation (1 min at 95 °C), denaturation (40 cycles of 95 °C for 10 s), annealing temperature range (60 °C for 5 s), and extension (72 °C for 10 s). For RT-qPCR analysis, three replicates(C17_1, C17_4, and C17_7) were taken, and the experiment was repeated thrice. We computed relative expression levels using the 2−ΔΔCt methods.

Statistical analysis

The D’Agostino-Pearson omnibus test was used to confirm that the data were normally distributed. Student’s t-test, one-way ANOVA, and two-way ANOVA were also applied to the data (for comparisons with more than one variant and across three or more groups). The numerous comparisons that followed the ANOVA were subjected to post hoc test with Bonferroni correction.

Synephrine contents in C. aurantium L.

Since synephrine is a major component of AFI, we measured the total levels of the synephrine at the different development stages of C. aurantium. The highest synephrine contents of the C17 fruits collected in May 2020 and 2021 were 7.64 ± 0.07 mg/g and 6.64 ± 0.38 mg/g, respectively (Fig. 1). Specifically, synephrine levels in the three accessions decreased to below 25% in the C17_4 stage in 2020 and decreased further gradually until C17_7 (Fig. 1A). The synephrine contents exhibited a decreasing trend in all eight stages in 2020 and 2021, but the decrease was more significant and occurred earlier in 2020 than in 2021. C17_1, C17_4 and C17_7 samples collected in 2020 were selected for transcriptome analysis to study the key genes and enzymes involved in synephrine biosynthesis.

Figure 1 The synephrine content at the eight developmental stages of Citrus aurantium L. fruits in 2020 and 2021.

(A) The line plot represents the trends of synephrine content in the eight developmental stages in 2020 and 2021. (B) The boxplot displays the total synephrine content in 2020 and 2021. The dotted line shows the average synephrine content in 2020 and 2021.

Expression landscape among the nine libraries of C. aurantium

Nine libraries were built and sequenced for the C. aurantium fruits collected at three developmental stages (C17_1, C17_4, and C17_7). The RNA Agilent 2100 bioanalyzer assay results of the C. aurantium and quality control statistics of the sequencing data were shown in Tables S1 and S2, respectively. We obtained 367.2 million raw reads from the nine libraries, which we then entered into the National Center for Biotechnology Information’s (NCBI) Sequence Read Archive (SRA) database (accession number: SAMN37179869, SAMN37179870, SAMN37179871, SAMN37179872, SAMN37179873, SAMN37179868, SAMN37179865, SAMN37179866, SAMN37179867). After filtering the low-quality sequences, we obtained an average of 40,796,727 clean reads per sample, with an average GC content of 44.14%. The average mapping rate of RNA-seq data was 91.65% (Table 1) when the C. sinensis genome was employed as a reference, suggesting that the data were suitable for additional study. For this reason, C. aurantium L.’s genome was chosen to be compared to that of C. sinensis.

Table 1 Summary of the sequencing quality of the nine RNA libraries of Citrus aurantium L.

Sample	Raw reads	Raw bases	Clean reads	Q30 (%)	GC content (%)	Total mapped (%)	
C17_1_1	40409858	6061478700	40409844	93.53	43.77	91.12	
C17_1_2	41068658	6160298700	41068646	93.79	43.84	90.98	
C17_1_3	40898712	6134806800	40898704	93.96	43.89	91.02	
C17_4_1	40961222	6144183300	40961218	93.62	44.06	92.04	
C17_4_2	40907714	6136157100	40907708	93.64	44.12	91.29	
C17_4_3	40919420	6137913000	40919416	93.81	44.17	92.27	
C17_7_1	41300182	6195027300	41300182	94	44.81	93.03	
C17_7_2	40393468	6059020200	40393464	94.12	44.35	91.07	
C17_7_3	40311366	6046704900	40311362	93.73	44.25	92.06	
Average	40796733	6119510000	40796727	93.8	44.14	91.65	

A total of 24,004 genes were detected, including 22,057 known genes and 1,947 novel genes. Moreover, 56,143 differentially expressed transcripts were obtained, including 38,749 known and 17,394 new transcripts. Based on the RNA-seq data’s fragments per kilobase per million reads (FPKM) (Fig. 2A), principal component analysis (PCA) was carried out for all nine samples. Three main components accounted for 0.82 of the variances overall (0.46, 0.26, and 0.10 for PC1, PC2, and PC3, respectively). The fact that the C17_1 sample clustered together suggests that the samples’ gene expression profiles were similar, which may be connected to the fruits’ strong production of secondary metabolites during the C17_1 stage. A heat map was used to display the correlation coefficients of the unigenes, with the highest expression level shown by the color red and the lowest expression level shown by the color blue (Fig. 2B). The expression patterns of the fruit libraries at the different growth stages were compared, and the results revealed that the C17_1, C17_4 and C17_7 libraries had similar colors and were classified into the same cluster. These results indicated that the samples were satisfactory for further analysis.

Figure 2 Principal component analysis (PCA) of the nine samples (A) and hierarchical clustering heat map showed correlation among samples based on fragments per kilobase per million reads FPKM of all unigenes (B).

Identification and functional analysis of DEGs during different fruit developmental stages

After applying specific criteria, namely Padj < 0.05 and —log2 fold change— > 1, we analyzed the functionality of DEGs at various fruit developmental stages. This analysis resulted in the discovery of 7309 DEGs (Table S3), obtained through pairwise comparisons conducted between different developmental stages (Fig. 3A). In the comparisons, the DEGs ranged from 690 to 3019, with a decrease at first and an increase later on as the time separation between the two stages increased. There was a greater number of DEGs between C17_1 and C17_7 than between C17_1 and C17_4, where there were 1995 DEGs. This trend was associated with relative differences in the synephrine content at different stages.

Figure 3 (A) Differentially expressed genes (DEGs) between the C171, C174 and C177 developmental stages of Citrus aurantium fruits.

Pairwise comparisons of the gene expression levels in the same sample at different stages. (B) The Venn diagram of the DEGs. Orange color represents C171 vs. C174, blue represents C171 vs C177, and green represents C174 vs C177.

By using the log2 (ratios) of the relative expression level, the DEGs were grouped using H-cluster, K-means, and sensitivity of method (SOM) methods. Multiple clusters were formed from the DEGs using various clustering algorithms. In Fig. S1, the relative expression of genes in a cluster at different development stages (depending on the expression level of the initial sample) is shown alongside the average of the relative expression of all the genes in the cluster at different development stages. The results showed that genes within the same cluster expressed themselves similarly at different stages of development. Venn diagram analysis indicated that the DEGs between C17_1 and C17_4, C17_4 and C17_7, and C17_1 and C17_7 were 1,380, 190, and 1,440, respectively. There were 481 common (intersection) DEGs for the three different stages of C. aurantium (Fig. 3B).

KEGG enrichment analysis of the DEGs

Understanding the transcriptome-level synthesis processes of certain metabolites, pertinent gene functions, and numerous gene connections is made possible through KEGG analysis. KEGG enrichment was investigated to link the DEGs to their functional categories. The analysis results indicated that 39 pathways had P-values < 0.05 among 132 enrichment pathways. Among them, the pathway labeled “Metabolism” had the highest level of enrichment, with 901 differentially expressed unigenes identified in 33 pathways. The metabolic processes of amino acids (tyrosine, phenylalanine, and tryptophan) and the production of secondary metabolites (phenylpropanoid, tropane, piperidine, pyridine alkaloid, isoquinoline alkaloid, indole alkaloid biosynthesis, and other alkaloids) were the main subjects of the significantly enriched KEGG pathways (Fig. 4 and Table S4). There were 120 differentially expressed unigenes in all the compared groups, and the number of DEGs in the C17_1 vs. C17_7 group, including 628 up-regulated and 541 down-regulated unigenes, was more than in the other groups (Table S5).

Figure 4 The rich factor represents all of the annotated genes in the pathway as a ratio of DEGs to all other genes.

Among all annotated genes, the rich factor represents the ratio of DEGs to all genes in the pathway. After repeated hypothesis testing and correction, the p-value is represented by the q-value, which has a range of [0,1]. The more significant the enrichment, the closer the q-value is to zero.

Wheaton & Stewart (1969) used chemical labeling experiments to demonstrate that the biosynthetic pathway of the alkaloid synephrine in citrus plants involves decarboxylation, N-methylation, and β-hydroxylation of tyrosine to synthesize synephrine. Compared to tyrosine and phenylalanine, tyramine are inefficient precursors for synephrine synthesis. Tryptophan, phenylalanine and tyrosine are aromatic amino acids synthesized by plants and microorganisms (Herrmann, 1995). The condensation reactions of the glycolysis/gluconeogenesis pathway produce PEP, while those of the pentose phosphate pathway produce E4P. DHAP synthesis from PEP and E4P is the first reaction step of the shikimate pathway. The shikimic acid pathway is a common process for producing aromatic amino acids, acting as an association between primary and secondary metabolism. Shikimic acid is a precursor for producing aromatic amino acids (Santos-Sánchez & Hernández-Carlos, 2019). The study’s highly enriched KEGG pathways were mostly associated with the following pathways: tyrosine, phenylalanine, tryptophan biosynthesis map00400; pentose phosphate pathway map00030 (Fig. S3), glycolysis/gluconeogenesis map00010 (Figs. S2; S4), and tyrosine metabolism map00350 (Fig. S5). The yellow background genes inside the red boxes in each metabolic pathway indicate significant upregulation of the pairwise compared samples at the different growth stages, while green background genes illustrate significant downregulation.

Identification and functional analysis of DEGs involved in synephrine biosynthesis

Based on the enriched KEGG pathways, literature review and differential expression analysis, 25 genes related to alkaloid synthesis were obtained, and their expression levels are shown in Table 2. In the glycolysis/gluconeogenesis pathway, enolase (Eno) [EC: 4.2.1.11] was significantly up-regulated in the C17_1 vs C17_7 comparison group and was encoded by Cs_ont_5g037680. Another transcript that was significantly up-regulated in the C17_1 vs. C17_4 comparison group was encoded by Cs_ont_9g024260. The ENO gene catalyzes the conversion of 2-phophoglycerate (2-PG) to PEP in mammalian, but these functions are rarely reported in plants (Yang et al., 2022).

Table 2 Expression patterns and functional analysis of the synephrine-related differentially expressed genes (DEGs).

Gene name	C17_1 vs C17_4	C17_4 vs C17_7	C17_1 vs C17_7	Sum	KO_name	
Cs_ont_8g007110	no—down	yes—up	yes—up	2	ADT, PDT	
Cs_ont_5g001740	yes—up	no—up	yes—up	2	aroA	
Cs_ont_1g028510	no—up	no—up	yes—up	1	aroB	
Cs_ont_9g003710	no—up	no—up	yes—up	1	aroDE, DHQ-SDH	
Cs_ont_5g009120	yes—down	yes—up	yes—up	3	aroDE, DHQ-SDH	
Cs_ont_3g017540	no—up	yes—up	yes—up	2	aroDE, DHQ-SDH	
Cs_ont_1g002190	no—up	no—up	yes—up	1	aroK, aroL	
Cs_ont_7g015990	yes—up	no—up	yes—up	2	DDC, TyDC1	
Cs_ont_1g000160	yes—up	no—up	yes—up	2	DDC, TyDC2	
Cs_ont_2g020700	yes—up	no—down	yes—up	2	DDC, TyDC3	
Cs_ont_1g000110	yes—up	no—up	yes—up	2	DDC, TyDC4	
Cs_ont_1g003410	no—up	no—up	yes—up	1	DDC, TyDC5	
Cs_ont_1g000140	yes—up	no—up	yes—up	2	DDC, TyDC6	
Cs_ont_3g005820	yes—down	yes—up	yes—down	3	DDC, TyDC7	
Cs_ont_5g001720	yes—up	no—up	yes—up	2	tktA, tktB	
Cs_ont_3g011210	yes—up	yes—up	yes—up	3	aroF, aroG, aroH	
Cs_ont_1g021460	no—up	yes—up	yes—up	2	aroF, aroG, aroH	
Cs_ont_5g037680	no—up	no—up	yes—up	1	ENO, eno	
Cs_ont_9g024260	yes—up	no—down	no—up	1	ENO, eno	
Cs_ont_4g003990	no—down	yes—down	yes—down	2	GOT1	
Cs_ont_6g013390	yes—down	yes—up	no—down	2	GOT2	
Cs_ont_2g022750	yes—up	no—down	yes—up	2	TAT	
Cs_ont_4g013970	no—down	yes—down	yes—down	2	TAT	
Cs_ont_7g002300	yes—up	no—down	no—up	1	TYRAAT	
Cs_ont_4g024260	no—down	yes—up	no—up	1	TYRAAT	

The three comparison groups had 13 genes significantly enriched in the phenylalanine, tyrosine and tryptophan biosynthetic pathways (map 00400) and the shikimate pathway. The DEGs encoding functional enzymes in the synthesis pathways of these three aromatic amino acids are shown in Fig. S3. The DEGs encode eight enzymes, of which trpA, trpB, trpD, and trpE mainly catalyze tryptophan synthesis and thus were not analyzed.

3-deoxy-7-phosphoheptenoate synthase, 3-deoxy-7-phosphoheptanoate synthase (AroF/G/H) [EC: 2.5.1.54] catalyze the condensation reaction of PEP and E4P to synthesize DHAP, the first reaction step of the shikimate pathway, and are encoded by Cs_ont_3g011210 and Cs_ont_1g021460. The Cs_ont_3g011210 was significantly up-regulated in three comparison groups (C17_1 vs C17_4, C17_1 vs. C17_7 and C17_4 vs C17_7), while Cs_ont_1g021460 was significantly up-regulated in two comparison groups (C17_1 vs C17_7 and C17_4 vs C17_7) [EC: 2.5.5.1.54]. 3-dehydroquinate dehydrogenase I (AroDE) [EC: 4.2.1.10] and shikimate dehydrogenase (DHQSDH) [EC: 1.1.1.25] are encoded by Cs_ont_5g009120, Cs_ont_9g003710, and Cs_ont_3g017540. Cs_ont_5g009120 was down-regulated in the C17_4 vs C17_7 comparison group, but up-regulated in the C17_1 vs C17_4 and C17_1 vs C17_7 comparison groups. Cs_ont_9g003710 was only up-regulated in the C17_1 vs C17_7, and Cs_ont_3g017540 up-regulated in the C17_4 vs C17_7 and C17_1 vs C17_7 comparison groups. 3-phosphoshikimate1-carboxyvinyltransfe-rase (AroA) [EC: 2.5.1.19] encoded by Cs_ont_5g001740 was significantly up-regulated in C17_1 vs C17_4 and C17_1 vs C17_7 comparison groups. AroB [EC: 4.2.3.4] encoded by Cs_ont_1g028510 was only significantly up-regulated in C17_1 vs C17_7 comparison groups. These enzymes were hypothesized to be involved in the biosynthesis of the tyrosine which is the precursor substance of phenylethylamine-type alkaloid synephrine.

Arogenate dehydrogenase (NADP+) (Tyraat) [EC: 1.3.1.78] is encoded by Cs_ont_7g002300 and Cs_ont_4g024260, which were significantly up-regulated in the C17_1 vs C17_4 and C17_4 vs C17_7 comparison groups, respectively. This enzyme catalyzes L-arogenate dehydrogenation to produce tyrosine, and its encoding gene may be associated acid orange alkaloid synephrine synthesis. Furthermore, arogenate dehydrase (ADT) and /prephenate dehydrase (PDT) [EC: 4.2.1.91, 4.2.1.51] are encoded by Cs_ont_8g007110, which was significantly up-regulated in the C17_1 vs C17_7 and C17_4 vs C17_7 comparison groups. These three genes may be associated with synthesizing the acid orange alkaloid synephrine. Tyrosine aminotransferase (TAT)[EC: 2.6.1.5] was encoded by Cs_ont_2g022750 and Cs_ont_4g013970, which catalyze the synthesis of phenylalanine and tyrosine, respectively. Cs_ont_2g022750 significantly up-regulated the C17_1 vs C17_4 and C17_1 vs C17_7 comparison groups, but Cs_ont_4g013970 significantly down-regulated the C17_1 vs C17_7 and C17_4 vs C17_7 comparison groups. In the tyrosine metabolic pathway (map 00350), tyrosine decarboxylated to synthesize tyramine, a potent precursor of the synthetic alkaloid synephrine. Aromatic-l-amino acid decarboxylase (TyDC)[EC: 4.1.1.28] was encoded by seven transcripts (Cs_ont_1g000110, Cs_ont_7g015990, Cs_ont_1g000140, Cs_ont_1g003410, Cs_ont_2g020700, Cs_ont_1g000160, and Cs_ont_3g005820). Cs_ont_3g005820 was significantly down-regulated, while the other six transcripts were significantly up-regulated in the C17_ 1 vs C17_ 7 and C17_ 1 vs C17_ 7 comparison groups. These findings may mean that TyDC gene plays an important role in the growth and development of trifoliate orange and synephrine synthesis.3.6. Weighted gene co-expression network construction and analysis

In order to get additional insight into the particular genes that are strongly linked to alkaloid biosynthesis, we employed weighted correlation network analysis (WGCNA) to examine the transcriptome changes during the fruit development of C. aurantium L WGCNA was carried out utilizing DEGs from pairwise comparisons between different accessions at the different developmental phases within the same sample. The co-expression network was built using 7,309 DEGs, with 6,072 of being classified into eight modules (the gray module was excluded). The number of genes varied between the modules, with the black module having the fewest (141) and the turquoise module having the most (1,963). We discovered a strong correlation between the synephrine content and the turquoise and yellow modules regarding the linkages between the modules and physiological characteristics (Fig. 5A). Additionally, we evaluated how well these modules correlated with gene expression profiles. The module membership (MM) and gene significance (GS) scores exhibited strong associations in these modules. Heat map of the correlation between the WGCNA module and phenotypes showed that modules in yellow and turquoise had a positive relationship with C17_1 (Fig. 5B, Fig. S6, Table S6). The results showed a substantial relationship between the genes in these modules and the synthesis of synephrine. By analyzing the correlation coefficient between the genes and phenotypes, we were able to identify 16 DEGs in the two major transcriptome screening modules (yellow and turquoise) (Table S7). These 16 genes were ADT, PDT (Cs_ont_8g007110), aroA (Cs_ont_5g001740), aroDE (Cs_ont_3g017540, Cs_ont_9g003710and Cs_ont_5g009120), aroF/G/H (Cs_ont_1g021460 and Cs_ont_3g011210), aroK, aroL(Cs_ont_1g002190), DDC/TyDC (Cs_ont_1g000160, Cs_ont_2g020700 and Cs_ont_7g015990), ENO (Cs_ont_5g037680, Cs_ont_9g024260), TAT (Cs_ont_2g022750), and TYRAAT (Cs_ont_4g024260 and Cs_ont_7g002300). According to these findings, the production of synephrine maybe regulated by them in C. aurantium L.

Figure 5 The nine samples’ weighted correlation network analysis (WGCNA) network analysis demonstrates the relationship between the module’s phenotypes and genes.

(A) WGCNA module identification diagram. (B) Heat map of the correlation between the WGCNA module and phenotypes. The correlation between the modules and the developmental period is displayed, with the abscissa representing different developmental periods and the ordinate representing different modules. Blue indicates a negative correlation and red indicates a positive correlation.

Variance across the comparison groups

Plotting each gene’s expression is necessary to observe and evaluate transcriptional changes at the gene level. To determine the variance of the genes in each group, we employed ANOVA. We used our dataset to perform an ANOVA with three replicates for each of the three categories and for the most and least correlated groups (n = 2). Interestingly, some key genes shown in Table S7 were considered significantly differentially expressed in each group. After the group size was reduced to the two most correlated samples, the statistical significance of their differential expression vanished. The P-value results indicated that ten genes (ADT, PDT Cs_ont_8g007110, aroDE Cs_ont_3g017540 and Cs_ont_5g009120, DDC/TyDC Cs_ont_3g005820, ENO Cs_ont_9g024260, GOT1Cs_ont_4g003990, GOT2Cs_ont_6g013390, TAT Cs_ont_4g013970, TYRAAT Cs_ont_4g024260 and Cs_ont_7g002300) showedno significant variation within or between groups. Only 15 genes (aroA (Cs_ont_5g001740), aroB (Cs_ont_1g028510), aroDE (Cs_ont_9g003710), aro F/G/H (Cs_ont_1g021460 and Cs_ont_3g011210), aroK (Cs_ont_1g002190), TyDC(Cs_ont_1g000160, Cs_ont_2g020700, Cs_ont_3g005820, Cs_ont_7g015990, Cs_ont_1g000140, Cs_ont_1g000110 and Cs_ont_1g003410), tktA/B (Cs_ont_5g001720), ENO (Cs_ont_5g037680), and TAT(Cs_ont_2g022750)) showed significant variation within or between groups (Fig. 6). Specifically, although ADT/PDT (Cs_ont_8g007110), aroDE (Cs_ont_3g017540 and Cs_ont_5g009120), ENO (Cs_ont_9g024260) and TYRAAT (Cs_ont_7g002300 and Cs_ont_4g024260) were identified in the two major transcriptome screening modules (yellow and turquoise), they were absent from the “least correlated” DEG list (Table S7).

Figure 6 Multiple group differences test of individual gene analysis using one-way ANOVA.

The x-axis represents the different grouping categories, and the y-axis represents the average expression of a certain gene/transcript in each group. * 0.01 < P ≤ 0.05; ** 0.001 < P ≤ 0.01; *** P ≤ 0.001.

Validation of RNA-Seq data

To validate the accuracy of the RNA-seq data, we selected six DEGs (ENO(Cs_ont_5g037680), aroA(Cs_ont_5g001740), AroF/G/H(Cs_ont_3g011210), TyDC1(Cs_ont_7g015990), TyDC2(Cs_ont_1g000160) and TyDC3(Cs_ont_2g020700) at random and analyzed their expression levels using RT-qPCR (Fig. 7). Table S8 gives the specific primers for these genes. All of these unigenes showed similar expression patterns to those found in the RNA-Seq analysis. Hence, high repeatability and reliability were demonstrated by our finished RNA-seq study results, which would be helpful for future research focusing on important genes involved in synephrine accumulation in AFI.

Figure 7 (A–F) The relative expression levels of nine selected DEGs were compared by RNA-seq and qRT-PCR.

The line chart shows the gene expression level from the transcriptome (FPKM).

Discussion

In the present study, we measured synephrine, an alkaloid and the contents exhibited a decreasing trend in all eight stages of C. aurantium. Alkaloids are a major class of secondary plant metabolites demonstrating vital pharmacological effects that benefit human health and regulate plant physiological functions. Although there are many studies on the biosynthetic regulation of alkaloids (Bui, Rodríguez-López & Dang, 2023; Fan et al., 2021), little is known about alkaloid biosynthesis in sour orange. Wheaton & Stewart (1969) evaluated chemical-labeled transfer among phenolic amines, and suggested tyramine →N-methyltyramine →synephrine is most likely the synephrine pathway in “Cleopatra” Mandarin. For synephrine and other phenolic amines, tyramine was the most potent precursor, while tyrosine was significantly less effective. First, we need to understand the synthesis of tyrosine, which is an aromatic amino acid. The aromatic pathway interfaces with carbohydrate metabolism at the reaction catalyzed by 3deoxy-D-arabino-heptulosonate 7-phosphate (DAHP) synthase, and the condensation of erythrose-4-phosphate and PEP forms the 7-carbon sugar, DAHP. There is growing evidence to show that separate biochemical pathways of aromatic biosynthesis may exist in the spatially separated microenvironments of the plastid and cytosolic compartments. Separate enzyme systems of carbohydrate metabolism exist in the cytosol and in plastids that would be able to generate erythrose-4phosphate and PEP for entry into aromatic biosynthesis (Jensen, 1986).

In this study, the KEGG analysis results indicated that 39 pathways had P-values < 0.05 among 132 enrichment pathways, and the highly enriched KEGG pathways were mostly associated with tyrosine, phenylalanine, tryptophan biosynthesis map00400; pentose phosphate pathway map00030; glycolysis/gluconeogenesis map00010, and tyrosine metabolism map00350. Most of the pathways analyzed in this study were consistent with those obtained in previous studies, which provided some reference value for the screening and functional research of core genes involved into synephrine biosynthesis. A total of 41 unigenes were identified, among which 15 were enriched in glycolysis/gluconeogenesis and pentose phosphate biosynthesis; 17 were related to phenylalanine, tyrosine, and tryptophan biosynthesis; and nine were related to tyrosine metabolism. 20 of the 41 unigenes belonged to the two key modules (yellow and turquoise). Most synephrine synthesis-related genes had increased expression levels at C17_1 (Table S7). The two modules were intimately linked to the biosynthesis of synephrine, and the DEGs(aroA (Cs_ont_5g001740), aroDE (Cs_ont_9g003710and Cs_ont_5g009120), aroF/G/H (Cs_ont_1g021460 and Cs_ont_3g011210), aroK, aroL(Cs_ont_1g002190), DDC/TyDC (Cs_ont_1g000160, Cs_ont_2g020700 and Cs_ont_7g015990), ENO(Cs_ont_5g037680,) and TAT (Cs_ont_2g022750),) related to synephrine found in both modules were chosen as potential candidate genes for further study. By combining all the analysis results, we constructed a model for synephrine production in sour orange fruits by analyzing the important genes associated with the process in the two key modules (Fig. 8).

Figure 8 Synephrine biosynthesis pathways, including glycolysis/gluconeogenesis, tyrosine metabolism, phenylalanine biosynthesis, and tyrosine biosynthesis, are depicted using diagrams.

Pathway-related expression patterns of differentially expressed genes (DEGs). Color indicates the fragments per kilobase per million reads (FPKM) scale, with red denoting high expression and blue denoting low expression.

It is worth noting that tyramine was the most potent precursor for the synephrine. Tyramine was produced by aromatic-l-amino acid decarboxylase (DDC) or tyrosine decarboxylase (TyDC), which in turn synthesizes alkaloids (Wang et al., 2022). TYDCs have attracted considerable attention because of their role in the biosynthesis of pharmaceutically important monoterpenoid indole alkaloids and benzylisoquinoline alkaloids, respectively (Facchini, Huber-Allanach & Tari, 2000). TYDCs are pyridoxal-50-phosphate (PLP)-dependent enzymes that decarboxylate tyrosine to yield CO2 and tyramine. The first plant TYDC was isolated from was barley (Hordeum vulgare) root (Gallon & Butt, 1971), and several plant TYDCs have been cloned since then Bartley, Breksa 3rd & Ishida (2010); Hu & Zhang (2021). The study findings suggested that TYDCs play important roles in metabolism (Facchini, Huber-Allanach & Tari, 2000; Swiedrych, Stachowiak & Szopa, 2004), stress responses (Gao et al., 2021) and plant defense (Shen et al., 2021). Considering that ethylamine b-hydroxylases, including dopamine b-hydroxylase [E.C. 1.14.17.1], possess stringent substrate specificities (Levin & Kaufman, 1961), the enzymes likely to be responsible for conversion of tyrosine to synephrine are tyrosine decarboxylase TYDCs.

In this study, we obtained 12 TyDC genes, seven of which were differentially expressed in different growth stages of sour orange. TyDC 1-6 was positively correlated with synephrine content, while TyDC 7 was negatively correlated with synephrine content. These findings may mean that the TyDC gene plays an important role in the growth and development of trifoliate orange and synephrine synthesis. The qPCR validation also indicated that the expression levels of TyDC1, 2 and 3 were highest in C17_1, with significantly higher the synephrine content than in other samples. However, the biosynthetic pathway of synephrine in sour orange needs further investigation. Further research and experimental validation may be needed to further understand the functions and regulatory mechanisms of these genes.

Although we determined some genes related to synephrine synthesis, there are still many top DEGs with unknown functions, which maybe participate in the synthesis of synephrine. It is necessary to conduct more in-depth research and analysis of these genes, so as to enrich the molecular pathway of synthesis of synephrine in C. aurantium, and provide the basis for breeding high-yielding synephrine plants.

Conclusion

The content of synephrine in lime alkaloid was detected based on the HPCE method for two consecutive years (2020 and 2021), and it was found that the content of synephrine showed a significant declining trend with the development of fruit. We sequenced and analyzed C. aurantium L. fruits from three samples (C17_1, C17_4, and C17_7) at different developmental stages to identify the molecular regulation and key genes involved in synephrine synthesis. The results showed that 25 genes among these KEGG pathways may be related to synephrine synthesis. WGCNA analysis and adoption variance across the groups suggested that 16 genes might play a crucial role in synephrine synthesis and should therefore be further analyzed. Analysis of variance across the comparison groups, showed that only 11 genes were included because of the high correlated samples” increased intragroup variability, 16 genes were present in all three datasets, and they were absent from the “least correlated” DEG list. Therefore, the results suggested that 11 genes (aroA (Cs_ont_5g001740), aroDE (Cs_ont_9g003710), aroF/G/H (Cs_ont_1g021460 and Cs_ont_3g011210), aroK, arol (Cs_ont_1g002190), DDC/TyDC (Cs_ont_1g000160, Cs_ont_2g020700 and Cs_ont_7g015990), ENO (Cs_ont_5g037680), TAT (Cs_ont_2g022750), and TYRAAT (Cs_ont_7g002300)) might play a crucial role in synephrine synthesis and should therefore be further analyzed. Our research collected large-scale and thorough transcriptome data for synephrine biosynthesis in C. aurantium L. fruits, defining the synephrine production pattern during fruit development. The candidate genes for synephrine production in sour oranges were identified. These findings could be helpful for selection or genetic modification targeted at raising the sour orange’s synephrine content.

Supplemental Information

Supplemental Information 1 Raw data for Fig. 1

Supplemental Information 2 Raw data for Fig. 7

Supplemental Information 3 Supplementary Tables

Supplemental Information 4 Supplementary Figures

We thank Kanehisa Laboratories for licensing the KEGG pathway map images copyright (map00030; map00400; map00010; map00350).

Additional Information and Declarations

Competing Interests

Author Contributions

Data Availability

The authors declare there are no competing interests.

Can Zhong conceived and designed the experiments, analyzed the data, prepared figures and/or tables, authored or reviewed drafts of the article, and approved the final draft.

Xitao Yang performed the experiments, prepared figures and/or tables, and approved the final draft.

Juan Niu conceived and designed the experiments, analyzed the data, authored or reviewed drafts of the article, and approved the final draft.

Xin Zhou analyzed the data, authored or reviewed drafts of the article, and approved the final draft.

Jiahao Zhou performed the experiments, prepared figures and/or tables, and approved the final draft.

Gen Pan performed the experiments, analyzed the data, prepared figures and/or tables, and approved the final draft.

Zhimin Sun conceived and designed the experiments, prepared figures and/or tables, and approved the final draft.

Jianhua Chen conceived and designed the experiments, prepared figures and/or tables, and approved the final draft.

Ke Cao conceived and designed the experiments, authored or reviewed drafts of the article, and approved the final draft.

Mingbao Luan conceived and designed the experiments, authored or reviewed drafts of the article, and approved the final draft.

The following information was supplied regarding data availability:

The sequences are available at National Center for Biotechnology Information’s (NCBI) Sequence Read Archive (SRA): SAMN37179869, SAMN37179870, SAMN37179871, SAMN37179872, SAMN37179873, SAMN37179868, SAMN37179865, SAMN37179866, SAMN37179867.

The raw data is available in the Supplemental Files.

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
