# Peer review of "Transcriptome analysis of Citrus Aurantium L. to study synephrine biosynthesis during developmental stages"

_PeerJ, doi:10.7717/peerj.17965_

## Round 0.1 · original submission · Major Revisions

Dear Dr. Zhong,

Thank you for your submission to PeerJ.

It is my opinion as the Academic Editor for your article - Transcriptome analysis and HPCE profiling of synephrine biosynthesis in Citrus Aurantium L. during its key developmental stages - that it requires a number of major changes.

You are advised to carefully go through all the comments and suggestions raised by the reviewers and modify your manuscript in accordance with their recommendations. Specifically, you will have to exercise utmost care in addressing the changes related to the experimental design, and procedures followed to carry out the experiments. Moreover, you need to ensure that there is proper coherence among different sections of the manuscript.

It is important to point out that your revised submission will undergo additional peer reviews in order to ensure that you have appropriately modified your manuscript.

Hope to receive the revised manuscript ASAP.

**Language Note:** PeerJ staff have identified that the English language needs to be improved. When you prepare your next revision, please either (i) have a colleague who is proficient in English and familiar with the subject matter review your manuscript, or (ii) contact a professional editing service to review your manuscript. PeerJ can provide language editing services - you can contact us at [email protected] for pricing (be sure to provide your manuscript number and title). – PeerJ Staff

Reviewer 1 ·

Basic reporting

The data provided in this study such as synephrine contents and transcriptome during fruit development would be valuable resources for further researches. However, the data presentation seems not to be efficient enough.

Experimental design

L70 What is C17? Is it a cultivar? Explain it. Since C17 is continuously used as the sample names, C17 should be meaningful enough.
L74 Did you use a single plant? The individuals would be good for biological replicates.

Validity of the findings

no comment

Additional comments

Major concerns
The abbreviation HPCE in title is not appropriate. Recommend to delete HPCE in the title. In addition, lower case “a” for “Citrus aurantium L.”
All figure legends should be reviewed again. The complete figure legends are hardly found in the manuscript. (It is incomplete in the pdf file)
L287-298 Current understanding about synephrine biosynthesis can be summarized in the introduction instead of discussion.
The DEG number itself is not so informative.

Minor revision
Recommend double line spacing for the revised manuscript.
Fig 2 legend for (B) is not appropriate. It would be correlation coefficients of whole unigene set.
L54 it is important
L64, 66, etc. Italicize C. aurantium
L82 r/min = rpm
L84 mmol/L = mM
L86 Describe about the external standard. How did you get that? Is it commercial, if so, please describe the manufacturer.
L130 Be consistent to make a space between number and the unit.
Fig 6. ‘One-way ANOVA’ on the top of the graphs is not necessary. Indicating significant difference with letters (a, b, c) as Fig 7.

Reviewer 2 ·

Basic reporting

The manuscript describes a transcriptomic analysis to identify the genes involved in the synephrine biosynthetic pathway. This work serves as a useful platform for future studies that can confirm the roles of the differentially expressed genes from this study. However, there is room for improvement, as outlined below. This manuscript therefore needs major revisions.

The manuscript should be revised to improve grammar. Some examples:
• italicize species names throughout manuscript
• some sentences need rewording. Consider having the manuscript proofread by a fluent English speaker.
• Several abbreviations are not explained at their first instance or not explained at all in the manuscript (e.g. Line 171: explain meaning of ‘SOM’).
• The reason for highlighting selected enzymes in yellow in Figs S2-S5 are not explained in the figure legend. While it is explained in the main text, it should be explained in the figure legend for better accessibility.
• Line 226: EC number incomplete
• Hyphenate ‘down-regulate’ and ‘up-regulate’

The introduction does not discuss previous work on the genes involved in synephrine biosynthesis (e.g. https://doi.org/10.1016/j.nbt.2010.04.003). The synephrine biosynthetic pathway in citrus species should be briefly discussed, which would help to highlight existing gaps that your study seeks to fill. Additionally, in Line 62, the creation of cDNA libraries is not an objective – it helps you to achieve your actual objective of further understanding the genes involved in synephrine biosynthesis.

Several of the figure legends are incomplete.

Experimental design

• More details are required in the methodology to ensure that these experiments are reproducible. For example, the link from which the genome assembly was downloaded should be provided; more details on the qRT-PCR experiments are needed – thermal cycler used? Temperature cycles used? Was PCR efficiency calculated for the primers? Did you design the primers, and if so, which program was used? Etc.

• You mentioned that you performed clustering of the DEGs in the ‘Results’ section, but you have not described doing this in your ‘Methods’ section.

• Insufficient detail of the WGCNA analysis makes it difficult to replicate. For example, were default parameters used? Either further details should be provided in the method, or your R code should be saved to a public repository (e.g. GitHub) and the link is made available.

Validity of the findings

• Figure 4: Some of the bubbles in the bubble plot seem to have a colour that reflects an adjusted p-value > 0.05, i.e. they are not statistically significant. Why are they included in the diagram? Why aren’t the adjusted p-values shown in Table S3?

• The scale independence and mean connectivity graphs you used to choose a soft threshold of 9 should be shown as a supplementary figure to justify your choice of soft threshold.

• Consider overlaying the RNA-seq expression values as a line graph on the qRT-PCR bar charts to visually show the similarity in expression data from the RNA-seq and qRT-PCR data (OR plotting a scatterplot of the log2FC values from RNA-seq and qRT-PCR to show correlation between expression values).

• Raw Ct values for qPCR data should be provided for reproducibility.

• The conclusion should be rewritten to answer the main objectives of the study and the future implications of the results. You have included information in your conclusion that is not relevant to your main objective (e.g. number of raw reads, comparisons of the numbers of DEGs at various time point comparisons, KEGG pathway identifiers etc.).

Reviewer 3 ·

Basic reporting

Minor rephrasing is requested throughout the document to enhance reading clarity. I have listed a few examples where rephrasing would be helpful. Please see the additional comments section.

The article structure including figures and tables is acceptable.

Experimental design

Transcriptome analysis and HPCE profiling of synephrine biosynthesis in Citrus Aurantium L. during its key developmental stages

Reviewer Summary:

Citrus Aurantium (bitter orange) is used in traditional Chinese medicine in the form of extracts from dried whole fruit. Synephrine is the primary phytochemical extracted and it is used widely in the food and pharmaceutical industries. While synephrine is found in other citrus fruits as well, this study focuses on C. aurantium and aims to identify the genes and pathways involved in the production of synephrine, and investigates the decrease in synephrine concentrations as the fruit matures.

The authors sampled fruits every 15 days beginning at 45 days after full blooming (DAFB), until 150 DAFB. They evaluated synephrine levels in each stage and identified three that had significant differences - C17_1, C17_4, and C17_7. They performed this evaluation on plants harvested in 2020 and 2021. They chose to use samples from 2020 for transcriptome analysis since they found that the decrease in synephrine production was more significant and occurred earlier in 2020 compared to the samples from 2021.

Sequence data was submitted to SRA and accessible under accession numbers SAMN37179865 - SAMN37179873

Verbatim from the draft:
"The primary objectives of the study were to (1) create C. aurantium L. mRNA libraries, (2) search for genes associated with synephrine biosynthesis that were expressed differently, and (3) build a correlation network for synephrine biosynthesis and important enzyme-encoding genes in the process."

The authors analyzed synephrine content using high-performance capillary electrophoresis, generated 9 cDNA libraries (3 growth stages, with 3 biological replicates each), sequenced the cDNA libraries and performed gene differential expression analysis, pathway analysis, and constructed co-expression networks. RT-qPCR was performed on 8 genes to validate the RNA-Seq data.

However, the authors have used C. sinensis reference genome for the transcriptome analysis.

Major Revisions Requested:

1. It is normally an accepted method to use a closely related and well-studied species as the reference genome when the genome or transcriptome of the species under study is not available. However, the authors should present the case for using C. sinensis as the reference genome in this study since the goals of this study are to learn about genes and pathways involved in synephrine production in C. aurantium, and how the concentration reduces with maturity of the C. aurantium L. fruit.

The authors should discuss the C. sinensis genes known to be involved in synephrine production, and the possibility of unique genes in C. aurantium L., that may be strongly associated with synephrine production but would be missed in this study. The alternative would be to de novo assemble the transcriptome of C. aurantium L., annotate it, and then use it for analysis of differentially expressed genes.

The corresponding author published a paper in 2022 titled "Transcriptome Analysis and HPLC Profiling of Flavonoid Biosynthesis in Citrus aurantium L. during Its Key Developmental Stages". In that study, 27 cDNA libraries were generated from 3 accessions at the same 3 growth interval stages. C. sinensis genome was used as the reference in that paper as well. Would it be possible to pool the libraries and generate de novo transcriptome for C. aurantium L. and use it for these analyses?

2. The Results section mentions "Assembly and annotation of the C. aurantium L. transcriptome". The section header suggests that de novo transcriptome assembly was produced for C. aurantium, and annotated.

Line 149: "A total of 24004 genes were detected, including 22057 known genes and 1947 novel genes."
Line 150: "56143 differentially expressed transcripts were obtained, including 38749 known and 17394 new transcripts".

Was a de novo transcriptome assembly generated? Or is this C. sinensis reference-based assembly? Please submit the method and the results from the transcriptome assembly and the annotation.

Validity of the findings

Major Revisions Requested:

3. Lines 203 - 204: "Based on the enriched KEGG pathways, literature review and differential expression analysis, 24 genes related to alkaloid synthesis were obtained, and their expression levels are shown in Table 2."
It would be very useful if a volcano plot was generated for stage 1 vs stage 7 and these 24 genes were marked on the plot.

4. It would be interesting and beneficial to the community if an analysis of the top differentially expressed genes (stage 1 vs stage 7) is presented. How many of these are unannotated? Were any of them unique to C. aurantium?

Additional comments

Minor revisions requested:

1. All figure legends appear truncated. This perhaps happened during the document conversion to pdf format. Please share the figure legends.

2. What is the source of the C. sinensis reference genome that was used?
- Provide the RefSeq or other database link to the genome used.
- Are the gene IDs, for example, Cs_ont_8g007110, from the reference C. sinensis genome?

3. Line 101: "HISAT2 (v2.1.0) with orientation mode to obtain unigenes" => What is "orientation mode"?

Lines 144 - 146: "The average mapping rate of RNA-seq data was 91.65% (Table 1) when the C. sinensis genome was employed as a reference."
What mapping parameters were used?

4. Rephrase the following for clarity:
Lines 73 - 76: "Three biological replicates were produced for each stage, with four fruits from the plant's northwest and southeast combined to create one biological replicate for each sample. The fruits that were obtained were divided in half. One half was used to analyze the synephrine content, and the other half was quickly frozen in liquid nitrogen and stored in an ultra-low freezer (-80°C) for transcriptome sequencing."

5. Please review the listed author affiliations.
- Inconsistencies in city, and province names.
- Affiliation #1 and #2 seem they are the same, but institute and academy are flipped
- Affiliation #1 and #4 are the same

6. Line 41. "AFI effectively treats gouty arthritis, cancer, and cardiovascular disorders when administered alone to eliminate phlegm". - Rephrase.... "...when administered alone to eliminate phlegm"

7. Line 45. "... and other healthcare sectors" change to "... healthcare and other sectors".

8. Abstract: "Transcriptome sequencing was used to examine the DEGs associated with synephrine accumulation and the connections between gene expression and synephrine content, which served as the foundation for creating synephrine-rich C. aurantium L." - Please rephrase for clarity.

9. Line 58. C17-4 => C17_4

10. Line 62. Italicize C. aurantium L. (Check this throughout the draft).

11. Lines 79 - 80. Is the sentence regarding statistical power calculation using RNASeqPower mistakenly placed here? The lines 78 - 87 describe calculating the synephrine content. Not RNASeq.

12. Line 94: "embryonic phases"?

13. Line 109: "KOBAS[19] analyzed the KEGG pathway on the Majorbio platform" => "KEGG pathway enrichment analysis was performed using KOBAS on the Majorbio platform".

Reviewer 4 ·

Basic reporting

In the manuscript “Transcriptome analysis and HPCE profiling of synephrine biosynthesis in Citrus Aurantium L. during its key developmental stages”, the authors investigated the genes and pathways involved in the developmental stages of Citrus Aurantium L. for potential targets for synephrine production. Improvements are needed.

a. The titles for the results sections should summarize the findings. Section 3.2 title section says “Assembly and annotation of C.aurantium L. transcriptome”, is there genome assembly and annotation? The content in this section seems to be mapping the reads to the reference genome to create gene count table and identify the DEGs.
b. The results section needs further organization. Please move technical descriptions to the methods section and in-depth discussions in the discussion section. The sentences that described the figure legends should go to the legends.
c. The paper should focus on the most important message for the results section. It’s hard to follow when all the pathways were stacked in the results section. Please highlighted the results that are mostly related to addressing the research question. Please focus on discussing the biological significance of the findings for the results and conclusion sections.
d. The importance of studying synephrine should be mentioned in the abstract as the motivation of the study.
e. The limitations for the study should be discussed.

Experimental design

The method of the study needs clarification.
a. How were the three biological replicates determined? Are they from the same plant or three different plants? If only one plant was studied throughout the study, then this is issue for potential bias in the study.
b. Why different methods were used for the DEG analysis (Methods section 2.4)? Which analysis produced the results shown in the results section?
c. For WGCNA analysis, which DEG list was used for constructing the gene co-expression networks? What are the UGT gene? What are the yellow and turquoise module for their biological functions?
d. In KEGG analysis, which DEG set was used? Between which comparison?

Validity of the findings

1. Quality control information, such as number of DEGs and number of raw reads don’t need to be mentioned in the abstract and conclusion. Please focus on summarizing the biological significance of the findings for the conclusion sections.
2. What does the “C17” stand for? These sample notations don’t need to appear in conclusions.

Additional comments

1. Please clarify the definition of “unigenes” in this study?
2. Line 153-155, why C17_1 sample cluster together are connected to the production of secondary metabolites?
3. Figure 6, the gene symbols should show instead of the gene IDs.

---

## Round 0.2 · accepted · Accept

Dear Dr. Zhong

Thank you for your submission to PeerJ.

I am writing to inform you that your manuscript - Transcriptome analysis of Citrus Aurantium L. to study the molecular of synephrine biosynthesis during developmental stages - has been Accepted for publication.

Congratulations!

The Section Editor noted:

- In the title, "Transcriptome analysis of Citrus Aurantium L. to study the molecular of synephrine biosynthesis during developmental stages", what do you mean "the molecular"? I think that this shoud just be deleted to "Transcriptome analysis of Citrus Aurantium L. to study synephrine biosynthesis during developmental stages." Edit "Citrus aurantium L., sometimes known as "sour orange," is an important Chinese herb with young, immature fruits, or "zhishi," that are high in synephrine. Edit "Transcriptome sequencing was used to examine the DEGs associated with synephrine metabolism and served as the foundation for creating synephrine-rich C. aurantium L." "From the nine libraries of ??? (what were the libraries from?), the number of DEGs varied from 690 to 3019 in the paired group comparisons (paired with what?). And so on.

Reviewer 4 ·

Basic reporting

The response and revision have addressed all the comments sufficiently.

Experimental design

Necessary details were added to the revision to explain the methods.

Validity of the findings

The comments were addressed.